# Study on the Evolution Law of Overlying Strata Structure in Stope Based on "Space–Air–Ground" Integrated Monitoring Network and Discrete Element

**Yuanhao Zhu** [1] , **Yueguan Yan** [1,*], **Yanjun Zhang** [1] , **Wanqiu Zhang** [1], **Jiayuan Kong** [1] and **Anjin Dai** [2]

1   College of Geoscience and Surveying Engineering, China University of Mining and Technology—Beijing, Beijing 100083, China; bqt2100204050@student.cumtb.edu.cn (Y.Z.); bqt2100204049@student.cumtb.edu.cn (Y.Z.); bqt2200204006t@student.cumtb.edu.cn (W.Z.); bqt2200204045@student.cumtb.edu.cn (J.K.)
2   College of Land Science and Technology, China Agricultural University, Beijing 100083, China; daianjin@cau.edu.cn
*   Correspondence: yanyuanguan@cumtb.edu.cn

**Abstract:** The geological environmental damage caused by coal mining has become a hot issue in current research. Especially in the western mining area, the size of the mining working face is large, the mining intensity is high, while the surface movement and deformation are more intense and wider. Therefore, it is necessary to effectively monitor the surface using appropriate means and carrying out research on the overlying strata structure of the stope. In this paper, by using advantages of various subsidence monitoring technologies and the technical framework of the Internet of Things (IoT), a "space–air–ground" integrated collaborative monitoring network is constructed. The evolution law of overlying strata structure is studied based on discrete element simulations and theoretical analysis. Furthermore, a discrete element mechanical parameter inversion method is proposed. The main results, using numerical simulations, are as follows: The mean square error of monitoring surface subsidence is 33.2 mm, the mean square error of mechanical parameter inversion is 13.4 mm, and relative error is as low as 3.8%. The surface subsidence law of adjacent mining under different working face widths and interval coal pillar widths is revealed. The Boltzmann function model of surface subsidence ratio changing with width–depth ratio and the calculation formula of width reduction coefficient of adjacent mining working face are inverted. The critical failure width of the interval coal pillar is determined as 20.5 m. Based on the theory of "arch–beam" structure and numerical simulation results, the overlying strata structure model of adjacent mining in the mining area is constructed. The research results can provide technical support or theoretical reference for mining damage monitoring, subsidence control, and prediction in western mines.

**Keywords:** mining subsidence; "space–air–ground" integrated monitoring network; discrete element simulation; adjacent working face; stope overlying strata structure

## 1. Introduction

As an essential source of coal resources in China, coal mining in the western mining area has caused serious geological and environmental damage problems while meeting energy demands. Especially in recent years, with the rapid development of coal mining technology, large-scale, rapid, and high-intensity mining has become the norm. The mining area is mostly characterized by the continuous mining of multiple large-size (generally 150 m–400 m) adjacent working faces. The law of overlying strata and surface movement caused by mining is more complex than that of a single working face. Moreover, the surface movement and deformation show the characteristics of fast subsidence, large deformation, and wide influence range, which also brings specific difficulties and various challenges to the surface monitoring [1].

The monitoring technology of a mining area mainly includes conventional surface monitoring technology, unmanned aerial vehicle (UAV) photogrammetry technology, and InSAR (interferometric synthetic aperture radar) monitoring technology. Among them, conventional surface monitoring technology arranges "point–line" observation stations on the main section of the strike or inclination of the working face and carries out conventional measurements such as GNSS (global navigation satellite system), traverse, and/or leveling [2]. The observation accuracy of this technology is high; however it has the disadvantages of high cost, heavy workload, easy damage to measuring points, and ease of being limited by topography. Therefore, meeting the needs of surface damage monitoring for high-intensity and large-scale mining in the western mining area is challenging.

The application of UAV photogrammetry in mining areas has a long history. This technology was originally used for topographic mapping [3]. D-InSAR (differential interferometric synthetic aperture radar) is a technology that uses the phase information of synthetic aperture radar complex images to obtain surface subsidence information. It has the advantages of monitoring all times of day, over a large area and at high precision. Similarly, the structure of overlying strata is key to clarifying the law and mechanism of mining subsidence. The existing literature has discussed various proposals based on field measurement, numerical simulations, and theoretical analysis. In terms of monitoring the surface deformation of mining areas, relevant research works are deliberated in Section 2.

In summary, the existing monitoring methods have their advantages and limitations, and we noted that a single technology is difficult to meet the western mining area's economical, efficient, ecological, and comprehensive monitoring needs. Therefore, we believe that the "space–air–ground" network, when integrated with monitoring networks, might fulfill the aims of integrating the various advantages of each technology. Furthermore, its monitoring results can provide important basic data for the numerical simulation of coal mining overlying strata movement. The stope's overlying strata structure can link the mining of the working face with surface movement, mainly including the arch and beam/plate theories. However, the current research on the evolution of the overlying strata structure is mostly focused on a single working face. There is a lack of in-depth discussion on the overlying strata structure of adjacent multi-working face mining, and this is rarely addressed in the existing state-of-the-art literature.

Therefore, this paper takes 3104 and 3106 working faces in a western mining area as background conditions and constructs an integrated "space–air–ground" monitoring network. Subsequently, it inverts discrete element mechanical parameters based on collaborative monitoring data and establishes a discrete element calculation model for adjacent mining of the working faces. Finally, using real datasets and certain numerical simulations, we explore the evolution law of the "arch–beam" structure of adjacent mining overlying strata of working faces. The major contributions of this paper are as follows:

- Using the advantages of various subsidence monitoring technologies and the technical framework of the Internet of Things (IoT), a "space–air–ground" integrated collaborative monitoring network is constructed.
- The evolution law of overlying strata structure is studied based on discrete element simulations and theoretical analysis.
- A discrete element mechanical parameter inversion method based on the "space–air–ground" integrated monitoring network is proposed.
- Based on the theory of "arch–beam" structure and numerical simulations results, the overlying strata structure model of adjacent mining in the mining area is constructed and evaluated using real datasets.

The rest of this paper is organized as follows: A brief discussion of various existing state-of-the-art approaches is deliberated in Section 2. In Section 3, we offer an overview of the study area and discuss the available datasets. In Section 4, we elaborate the proposed research methodology and suggest a numerical model. The proposed model is evaluated using numerical experiments and several assumptions in Section 5. We discuss the obtained

results and their analysis in Section 6. Finally, we conclude this paper in Section 7 and discuss few directions for future research.

## 2. Related Work

In terms of UAV monitoring, relevant scholars have also carried out a number of studies. In previous studies [4,5], UAV aerial survey technology was used to obtain DEM (digital elevation model) data on the surface of coal mining subsidence areas, which demonstrates and verifies that the data accuracy could reach to the centimeter level. Paweł et al. [6] used UAV photogrammetry technology to monitor the surface discontinuous deformation of the mining area and verified the practicability of the UAV in the domain of mining area monitoring. Ge et al. [7] used drones to observe the surface of the Tahmoor mining area in New South Wales, Australia, and plotted the subsidence curve of the surface of the mining area. Zhou et al. [8,9] used the UAV photogrammetry approach to monitor the surface subsidence of a coal mining area, inverted the subsidence prediction parameters, and verified that the accuracy of the subsidence basin was 81 mm. Puniach et al. [10] obtained high-resolution digital orthophoto images before and after surface deformation in the mining area using UAV photogrammetry. Furthermore, a weighted, normalized cross-correlation algorithm was used to constrain the matching results, and the obtained horizontal movement was compared with ground 3D laser observations. The authors observed that the accuracy can reach up to 1–2 pixels. Dai et al. [11] used UAV technology to obtain orthophoto images of tailings dams and monitored the maximum subsidence range within 0.16 m. However, UAV technology is still limited by factors such as cost and accuracy in mine monitoring.

In terms of D-InSAR monitoring, Gabriel et al. [12] first used D-InSAR technology to separate the deformation phase from the terrain phase in the interferometric phase and confirmed that the monitoring accuracy of surface deformation can reach centimeter or even millimeter level. In 1996, Carnec et al. [13] first used the D-InSAR method to monitor the surface subsidence of the mining area near Gardanne; the maximum subsidence value obtained was 42 mm, and the root mean square error of monitoring was approximately 459 mm. Moreover, it was found that the differential SAR interferometry was not suitable for monitoring large gradient deformation areas in a short time. Similarly, Yang et al. [14–16] monitored the surface deformation caused by mining in the mining area based on monorail InSAR, time series InSAR, and the combination of InSAR technology and leveling. The authors verified that the root mean square error was between the predicted surface subsidence value and the InSAR monitoring is 2.15 cm and analyzed its subsidence law. Zhang et al. [17] proposed the fusion of "D-InSAR measurement (space)" and "radon monitoring (ground)" to monitor surface mining cracks in mining areas. However, due to the influence of space–time decoherence, atmospheric delay, and orbital error, D-InSAR technology could reliably obtain the large deformation in the central area of the subsidence basin.

Scholars have made some achievements in the study of overlying strata structure; these proposals and the obtained results are largely related to a single working face and multiple working faces are relatively unexplored. In addition to this shortcoming, all previous studies have collectively shown that there is vertical zoning and horizontal zoning in the mining strata of the working face [18]. In terms of rock mechanical structure, cantilever beam theory, pressure arch theory [19,20], hinged rock block hypothesis [21], and key stratum and voussoir beam theory [22,23] are the main contributions. Wu et al. [24] proposed the supporting plate theory in strip or room-and-pillar mining prediction theory. Aiming at the problem of mining pressure behavior in adjacent mining, Jiang [25] proposed four types of overlying strata spatial structure, i.e., (i) θ-shaped, (ii) O-shaped, (iii) S-shaped, and (iv) C-shaped.

He et al. [26,27] proposed the dynamic evolution process of overlying strata "OX–F–T" structure, and provided the mechanical conditions for the instability of key stratum in adjacent goafs. Yang et al. [28] studied the stability of a goaf roadway in adjacent working

faces of the same coal seam. Based on the key stratum theory and 3DEC numerical simulation, Yu et al. [29] put forward their observations under the conditions of fully mechanized top-coal caving mining with large mining depth. The authors observed that with the increase in the total quantity of working face mining, the surface experiences the breaking form of the key stratum in the extremely insufficient–insufficient–sufficient process, which has a direct impact on surface movement and deformation.

## 3. Overview of the Study Area and Datasets

### 3.1. Study Area

The research area is located in a coalmine in Ordos City, Inner Mongolia Autonomous Region. The area includes two adjacent mining faces, 3104 and 3106. The inclination length of 3104 and 3106 is 300 m; the strike lengths are 1362 m and 1552 m, respectively; and the width of the interval coal pillar of the working face is 30 m. The mining durations for both mining faces of 3104 and 3106 are from March 2018 to October 2018 and from November 2018 to May 2019, respectively. The position and advancing direction of the working face are shown in Figure 1. The working face belongs to 3–1 # coal seam, with an average mining thickness of 5.2 m. The coal seam dip angle is 2°, which belongs to a near-horizontal coal seam, and the mining depth is about 420 m. The roof is managed using the whole caving method, and the lithology of the overlying strata is medium hard. The surface vegetation of the working face is sparse, mostly low shrubs, with a small number of buildings, but with no large lakes and rivers.

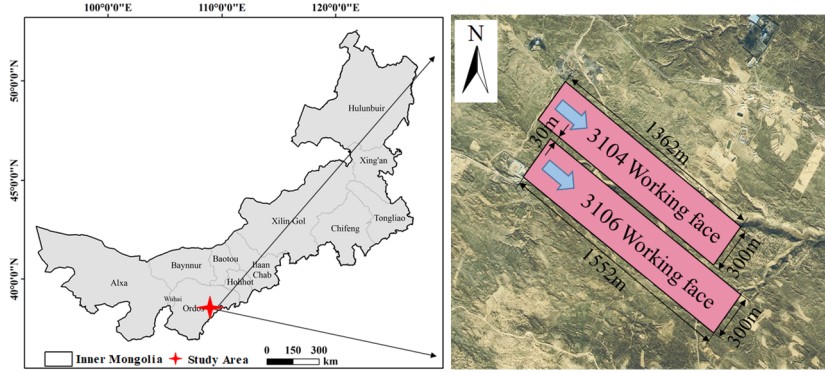

**Figure 1.** Working face position and advancing direction.

### 3.2. Datasets

The monitoring datasets that are used within this study area include InSAR, UAV, as well as GNSS data. Note that the latter dataset of surface observation stations is used for supplement and verification. The data source information of the study area is shown in Table 1. The UAV data collection in the study area was carried out on 19 July and 5 October 2018. The UAV flight platform adopts D2000 FEIMA intelligent aerial survey system, equipped with a visible light CMOS sensor. UAV manager software was used for route planning and parameter setting. The forward overlap parameter was set to 80%, and the side overlap parameter was set to 75%. Furthermore, the ground resolution of the image was designed to be 5 cm.

According to the observation time of the UAV, the InSAR data in the study area selected eight different scenes of Sentinel-1A IW SLC image data between 17 July 2018 and 9 October 2018. The data were launched by ESA in April 2014, equipped with a C-band radar having a bandwidth within the rage of 0–100 MHz. The interference data were VV polarization data with interference wide (IW), and the spatial resolution was 5 m × 20 m. The external DEM is the Space Shuttle Radar Terrain Mission (SRTM) data with a resolution of 30 m.

**Table 1.** Data source information of the study area.

| Data Source | Acquisition Date | Other Information |
|:---:|:---:|:---:|
| InSAR | 17 July 2018, 29 July 2018<br>10 August 2018, 22 August 2018<br>3 September 2018, 15 September 2018<br>27 September 2018, 9 October 2018 | Time baseline (12d)<br>Imaging mode (IW)<br>Spatial resolution (5 m × 20 m) |
| UAV | 19 July 2018<br>5 October 2018 | Sensor type (CMOS)<br>Sensor size (23.5 × 15.6 mm (aps-c))<br>Spatial resolution (5 cm) |
| GNSS | 19 July 2018, 5 October 2018 | – |

In the study area, in addition to the image control points of UAV photogrammetry, three observation stations were set up at the edge, inflection point, and maximum subsidence point of the subsidence basin. The coordinates and elevation information of each observation station were obtained using the GNSS-RTK measurement method. The GNSS measurement data of the study area were collected on 19 July and 5 October 2018.

## 4. Research Methods

### 4.1. "Space–Air–Ground" Integrated Monitoring Network

The "space–air–ground" integrated monitoring network coordinated using InSAR, UAV, and GNSS technologies was used in the study area. Furthermore, Internet of Things (IoT)-based monitoring systems are largely used for determining geotechnical information [30]. IoT systems supported by wireless networks can be used to gather, move, and process land displacement data in the mining environments. The monitoring network mainly includes four parts: (i) space monitoring module; (ii) sky monitoring module; (iii) ground monitoring module; and (iv) IoT-based data fusion framework. Among them, in the "space–air–ground" monitoring module, the observation accuracy of InSAR technology can reach the millimeter level [12]. The observation data can better describe the edge of the subsidence basin, but the central part of the basin is seriously decoherent. UAV technology can effectively identify large deformations in the middle of the basin, but the edge observation accuracy is low. Therefore, combining the advantages of both, D-InSAR technology is used to monitor the small deformation of the edge of land subsidence basin. Similarly, UAV technology is used to monitor large deformations of the center of the surface subsidence basin to obtain surface subsidence information comprehensively and accurately. GNSS technology not only assists UAV measurement, but also serves as an effective supplement and verification for "space–air" monitoring. It should be noted that the proposed IoT technology framework can provide connections between various monitoring modules using data sharing and optimize and reduce the amount of data by using data fusion and aggregation methods [31] to improve the efficiency and accuracy of surface movement monitoring, as shown in Figure 2.

The IoT technology framework proposed in this paper is used to collect, store, and manage multi-source monitoring data (including UAV data, InSAR data, and GNSS data). The entire environment is monitored using various sensors that collect different types of data. In fact, IoT devices represent a network of geotechnical surface sensors (as shown in Figure 2) that is responsible for collecting data without human intervention [30]. The devices are connected to a centralized internet-enabled computer that is used to store data. Subsequently, this computer could also be connected to a remote cloud to store the huge amount of gathered data more efficiently and securely. However, the monitoring data can be quite bulky and may contain data that represents similar measurements. Therefore, moving data across sensors and the cloud would be time consuming and create response issues. Therefore, in addition to data collection analysis, its functions mainly include data preprocessing and aggregation, data fusion, and accuracy evaluation. In the data preprocessing part, some monitoring data with large errors and outliers can be directly

eliminated. Furthermore, in the data aggregation phase, redundancy of multi-source data can be solved. This is due to the fact that a single area can be monitored using various IoT sensors; therefore, the original collected dataset might contain duplicate information. This duplicate information can create network issues, and learning from huge amounts of data requires a significant amount of training. We used the well-known concept of Euclidian distance to remove redundant data. In the data fusion part, based on the threshold range of surface subsidence monitored using UAV and InSAR, combined with the probability integral method, the effective monitoring of a surface subsidence basin in a mining area was realized. In the accuracy evaluation part, the validity and correctness of the monitoring data was accurately evaluated.

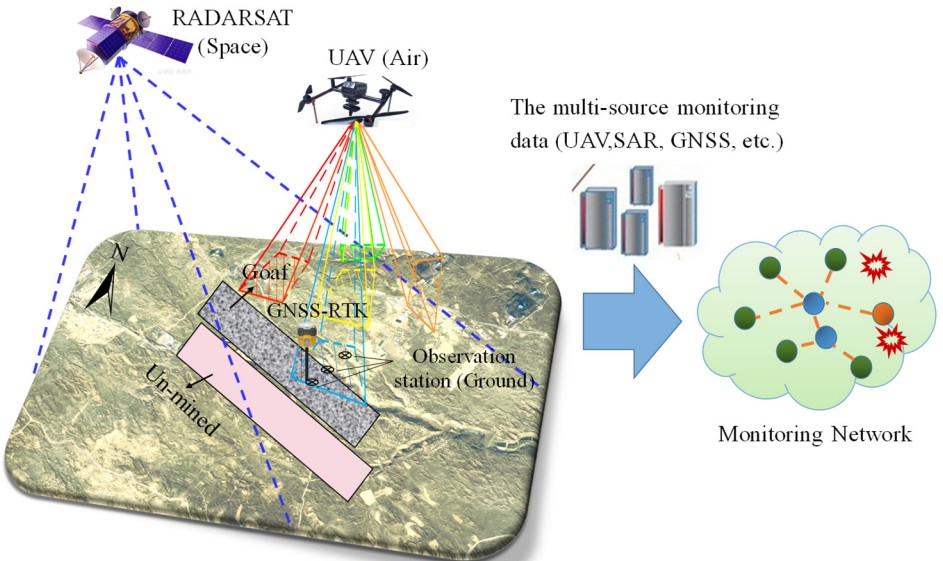

**Figure 2.** The "space–air–ground" integrated monitoring network.

### 4.1.1. UAV and InSAR Solution of Subsidence Basin

The basic principle of monitoring surface subsidence by the UAV photogrammetry is as follows: At any two time points during the mining period at the working face, the surface is observed using a UAV, and the digital elevation model (DEM) is solved at that time. Then, the surface subsidence basin of the monitoring area is obtained by subtracting the two DEMs. In this paper, based on the two-stage UAV tilt image data and the image control point data collected using GNSS-RTK, software programs ContextCapture and Pix4D Mapper were used to process the data. The main steps include image preprocessing, free aerial triangulation, constrained aerial triangulation, dense matching, and three-dimensional point cloud filtering based on a cloth simulation filtering algorithm [32,33]. Similarly, the filtered point cloud data were obtained using the inverse distance-weighted interpolation algorithm and the digital elevation model. Finally, we then constructed the surface subsidence basin, as shown in Figure 3a, where the maximum surface subsidence value is 2823 mm, which is close to the 2856 mm observed by GNSS. The inclined main section of the 3104 working face was selected to draw the subsidence curve. The point density is one every 30 m, and the point number is Q1–Q26, as shown in Figure 4.

Combined with the external DEM, SARscape software was used to process the data of 8 InSAR images to obtain the deformation map of 7 interference image pairs. D-InSAR technology was used to obtain the one-dimensional deformation of the radar line-of-sight (LOS), and then the LOS deformation was converted into vertical deformation. Finally, the deformation of each interference image pair was superimposed and calculated. In this way, the surface settlement of the 3104 working face observation time from 17 July 2018 to 9 October 2018 was obtained, as shown in Figure 3b. However, the maximum subsidence monitored using InSAR is only 112 mm, which is much smaller than the actual subsidence.

Therefore, when the InSAR monitored the large deformation gradient of the mining area, it was unable to accurately obtain the subsidence of the large deformation area due to the influence of time–space decoherence factors. The inclined main section of the working face was selected to draw the subsidence curve, as shown in Figure 4.

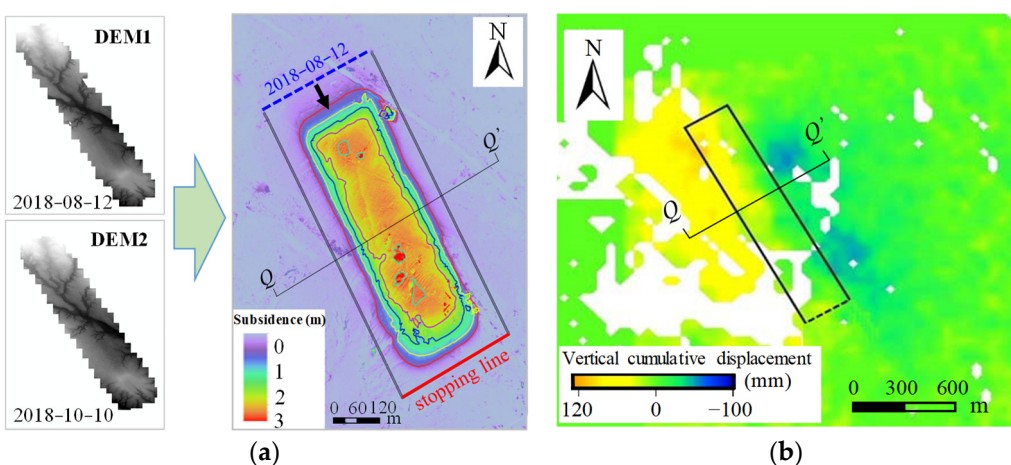

**Figure 3.** Subsidence basin solved using InSAR and UAV. (**a**) UAV solution of subsidence basin; (**b**) InSAR UAV solution of subsidence basin.

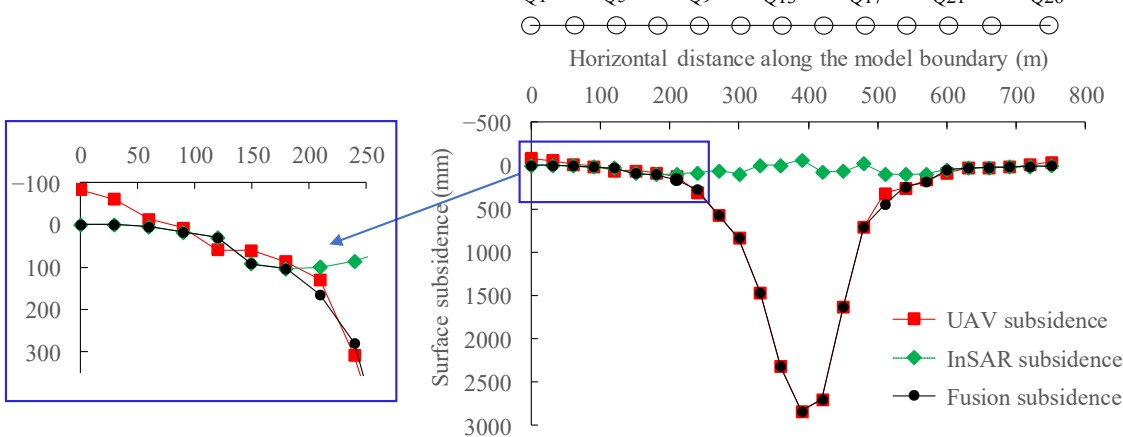

**Figure 4.** The UAV/InSAR fusion *subsidence basin*.

### 4.1.2. UAV/InSAR Fusion Subsidence Basin

Data fusion of UAV and InSAR is needed to calculate the threshold of the observation results of the two, and then determine whether the effective observation intervals of the two overlap or not. If they overlap, then they can be directly fused. However, if are not overlapped, then the UAV data are selected for large deformation while the InSAR data are selected for small deformation. Furthermore, the interval part in the middle can be fitted and interpolated using the probability integral function. The maximum deformation gradient that the InSAR can detect can be calculated using Formula (1) [34]:

$$D_{\max} = d_{\max} + 0.002(\gamma - 1) \tag{1}$$

In the above formula, $D_{\max}$ represents the maximum deformation gradient that the InSAR can monitor. Similarly, $\gamma$ is the coherence coefficient and its value range is between 0 and 1. The parameter $d_{\max}$ characterizes the maximum theoretical deformation gradient of the InSAR's monitoring, which is equal to the ratio of satellite sensor wavelength to 2 times the image resolution. For this 20 m resolution C-band Sentinel-1A image, when the coherence coefficient is less than 0.3, then the maximum detectable deformation gradient

of the D-InSAR is close to 0. Therefore, this value is used as the screening threshold for InSAR monitoring. The coherence coefficient of each monitoring point is extracted, and the coherence coefficient of the monitoring point between Q7 and Q21 is determined to be less than 0.3.

From the above observations, we concluded that the UAV can effectively monitor the large deformation area in the center of the subsidence basin, and its monitoring accuracy can reach the centimeter level. According to observations from the literature [35] combined with the theory of mining subsidence, the threshold for UAV observation is 0.16 Wmax, where Wmax denotes the measurement metric, showing the maximum subsidence value of the surface.

Therefore, the observation thresholds of UAV and InSAR are determined as 104 mm and 457 mm, respectively. For that reason, the InSAR observation data were used for Q7 points, and before, Q21, and after, the UAV observation data were selected between Q10 and Q18, and the remaining points were supplemented using the probability integral method fitting interpolation, as shown in Figure 4. It should be noted that the fusion results can lay a foundation for the determination of the physical and mechanical parameters of the overlying strata using discrete element simulations.

### 4.2. Discrete Element Numerical Simulation

#### 4.2.1. Numerical Model Establishment

The discrete element method is another powerful numerical method that has been largely used for analyzing the dynamics of material systems after the finite element method and computational fluid dynamics (CFD) are applied. This method regards the jointed rock mass as being composed of discrete rock blocks and joint surfaces between rock blocks, which can more realistically simulate the nonlinear large deformation characteristics of jointed rock mass. The key concept of the discrete element method is to assume the research object as a set of completely rigid or deformable blocks or spheres, and to simulate the force, displacement, and deformation of the assembly by defining the mechanical contact behavior between the blocks or spheres.

Like other numerical methods, the discrete element method has some limitations, such as the complexity of choosing mechanical parameters and the calculation of time step. In terms of the s election for mechanical parameters, the selection of different parameters affects the convergence of discrete element calculation. At the same time, the parameters are directly related to the correctness of the overlying strata movement process in a mining simulation. In terms of the determination of calculation time step, it should be kept in mind that, at present, the determination of the calculation time step is entirely based on the convergence of the mathematical equation's solution; however, it is not entirely related to the real-time concept.

Universal Distinct Element Code (UDEC) is a two-dimensional discrete element numerical simulation software developed by Itsac for discontinuous media. Its basic principle is based on Newton's second law and stress–displacement action law to simulate discontinuous media's stress and motion state under dynamic and static loads. UDEC software allows for large deformation of discrete blocks, which can slide, rotate, and fall off along discontinuous joint surfaces. It has become a vital calculation tool within the domain of mining subsidence research [36]. Due to its efficiency and wide use, UDEC software was selected for the discrete element numerical simulations in this paper. The geological and mining conditions in the study area were taken as the prototype. According to the distribution of the rock strata exposed by the mine boreholes, the model was simplified appropriately. We assumed that the model size is 1500 m × 450 m. The discrete unit block was divided into 18,550 units according to the actual development and distribution of the rock strata. The model contains two adjacent mining units of the 3104 and 3106 working faces, as shown in Figure 5. In the model, the strain softening model was used to describe the mechanical properties of the coal body, the Mohr–Coulomb model was used as the yield failure criterion of coal rock mass, and the Coulomb sliding model of surface contact was

selected as the joint. The upper boundary of the model is a free surface, the left and right boundaries limit the horizontal displacement, the lower boundary limits the horizontal and vertical displacement, and the initial stress field is the self-weight stress of the rock stratum. When the maximum unbalanced force was about 0.01% of the initial unbalanced force, then the model calculation was considered completed.

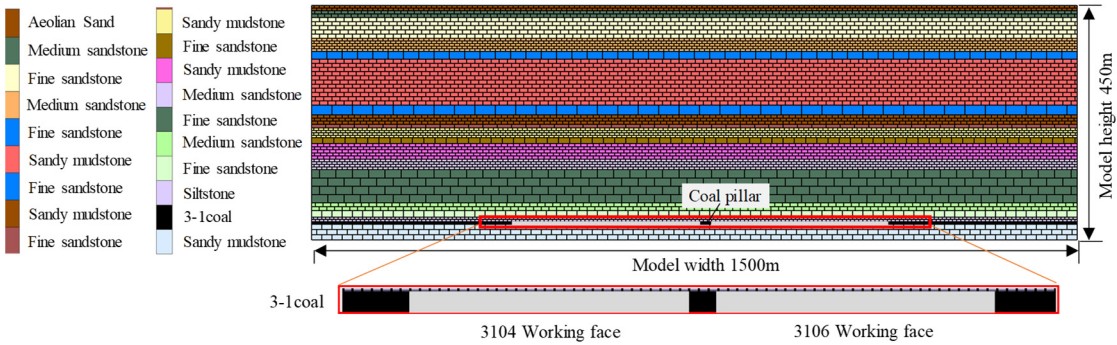

**Figure 5.** Discrete element calculation of mining in adjacent working face.

### 4.2.2. Inversion of Mechanical Parameters Based on "Space–Air–Ground" Integrated Monitoring Network

The traditional measured data used to invert the mechanical parameters are usually taken from the surface observation station data. However, it is evident from the existing state-of-the-art that this "point–line" observation method has various disadvantages such as easy loss of monitoring points, high cost, heavy workload, and susceptibility to terrain restrictions. The "space–air–ground" integrated monitoring network can overcome these shortcomings and help to obtain ground surface subsidence information more economically and efficiently.

The physical and mechanical parameters of coal and rock are determined on the basis of laboratory mechanics measurement and compared with the measured data of the working face surface. Combined with laboratory measurement and empirical parameters, using multiple sets of orthogonal tests, based on the similarity of surface deformation, the physical and mechanical parameters of coal rock mass were inverted using numerical simulation, as shown in Table 2. This set of parameters was used for the excavation of the adjacent 3106 working face and each simulation scheme.

The mining scale of the western mining area is significantly large, as is usual for the continuous excavation of multiple adjacent working faces. Adjacent mining involves many geological mining factors, such as coal thickness, mining depth, mining width, interval coal pillar width, strata lithology, and so on [37]. For specific projects, the geological conditions are fixed. The surface movement and overburden structure evolution are mainly affected by the mining width of the working face and the width of the interval coal pillar. In order to further study this law, note that the design of working face mining width was set to 150 m, 200 m, 250 m, 300 m, 400 m, and the interval coal pillar width was set to 15 m, 30 m, 50 m, 70 m, 90 m, 110 m.

**Table 2.** The mechanical and physical parameters of the coal rock in the UDEC model.

| Lithology | Thickness (m) | Density (kg·m$^{-3}$) | Bulk Modulus (GPa) | Shear Modulus (GPa) | Cohesion (MPa) | Friction Angle (°) | Tensile Strength (MPa) |
|---|---|---|---|---|---|---|---|
| Aeolian sand | 10 | 1580 | 0.67 | 0.53 | 0.2 | 10 | 0.12 |
| Medium sandstone | 13 | 2200 | 5.30 | 4.80 | 2.0 | 25 | 1.80 |
| Fine sandstone | 40 | 2500 | 5.80 | 4.30 | 1.1 | 45 | 1.53 |
| Medium sandstone | 25 | 2200 | 5.30 | 4.80 | 2.0 | 25 | 1.80 |
| Fine sandstone | 15 | 2500 | 5.80 | 1.37 | 1.1 | 45 | 1.53 |
| Sandy mudstone | 88 | 2390 | 2.27 | 1.17 | 3.3 | 28 | 1.72 |
| Fine sandstone | 19 | 2500 | 5.80 | 4.30 | 1.1 | 45 | 1.53 |
| Sandy mudstone | 18 | 2390 | 2.27 | 1.17 | 3.3 | 28 | 1.72 |
| Fine sandstone | 7 | 2500 | 5.80 | 4.30 | 1.1 | 45 | 1.53 |
| Sandy mudstone | 19 | 2390 | 2.27 | 1.17 | 3.3 | 28 | 1.72 |
| Fine sandstone | 11 | 2500 | 5.80 | 4.30 | 1.1 | 45 | 1.53 |
| Sandy mudstone | 30 | 2390 | 2.27 | 1.17 | 3.3 | 28 | 1.72 |
| Medium sandstone | 20 | 2200 | 5.30 | 4.80 | 2.0 | 25 | 1.80 |
| Fine sandstone | 64 | 2500 | 5.80 | 4.30 | 1.1 | 45 | 1.53 |
| Medium sandstone | 15 | 2200 | 5.30 | 4.80 | 2.0 | 25 | 1.80 |
| Fine sandstone | 12 | 2500 | 5.80 | 4.30 | 1.1 | 45 | 1.53 |
| Siltstone | 8.8 | 2400 | 3.00 | 3.97 | 3.8 | 43 | 1.50 |
| 3-1coal | 5.2 | 1480 | 1.33 | 5.62 | 0.9 | 35 | 0.72 |
| Sandy mudstone | 30 | 2390 | 2.27 | 1.17 | 3.3 | 28 | 1.72 |

## 5. Results and Analysis

### 5.1. Accuracy Assessment of "Space–Air–Ground" Integrated Monitoring Network

In order to verify the accuracy of the fused data, the GNSS observation results of the three monitoring points (Q3, Q9, Q14) at the center, inflection point, and the edge of the subsidence basin were, respectively, taken into account. Furthermore, these results were compared with the UAV/InSAR fusion benchmark results, and the root mean square error (RMSE) evaluation metric was calculated. The RMSE metric was computed using Formula (2). The monitoring data of each checkpoint are shown below in Table 3.

**Table 3.** Monitoring data of the checkpoints.

| Point Mark | UAV/InSAR Fusion Subsidence Value (mm) | GNSS Monitoring Subsidence Value (mm) |
|---|---|---|
| Q3 | 5 | 19 |
| Q9 | 325 | 280 |
| Q14 | 2823 | 2856 |

$$\sigma = \sqrt{\frac{\sum\limits_{i=1}^{n}(U_i - W_i)^2}{n}} \tag{2}$$

In the above formula, $U_i$ is the fusion subsidence value of UAV and InSAR at the $i$th monitoring point, $W_i$ is the GNSS-measured value at the $i$th monitoring point, and $n$ is the number of monitoring points. Substituting the corresponding data into Formula (2), the overall mean square error (MSE) of the fused subsidence basin was calculated and we noted its approximate value as 33.2 mm. Note that the MSE was also computed using the method provided in Formula (2).

The discrete element calculation model adopted the mechanical parameters based on the inversion of the "space–air–ground" integrated monitoring data. Furthermore, it simulated the excavation of the 3104 working face in the model, extracted the subsidence value of the surface point after mining, and then compared it with the fusion measured results, as shown below in Figure 6.

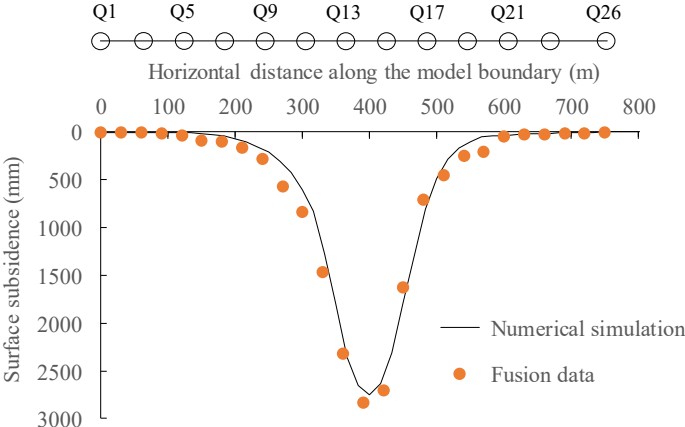

**Figure 6.** Comparison of surface fusion monitoring and numerical simulation subsidence results in 3104 working face mining.

Figure 6 shows that the simulated maximum surface subsidence value is approximately 2748 mm, which is very close to the monitored maximum subsidence value. It is also visible from the results that the overall mean square error (MSE) of the simulated subsidence and the measured surface fusion is 13.4 mm, and the relative error is approximately

3.8%, which verifies the reliability of the discrete element mechanical parameters and simulation calculation.

### 5.2. Analysis of Adjacent Mining Surface Subsidence Results

Under the condition of different interval coal pillar widths, the working face mining width changed, and the surface subsidence was calculated using discrete element simulation software. In order to facilitate analysis and expression, the left boundary of the interval coal pillar was used as the coordinate origin of the horizontal axis. The width of the working face is represented by a, and the width of the coal pillar is represented by b, as shown in Figure 7.

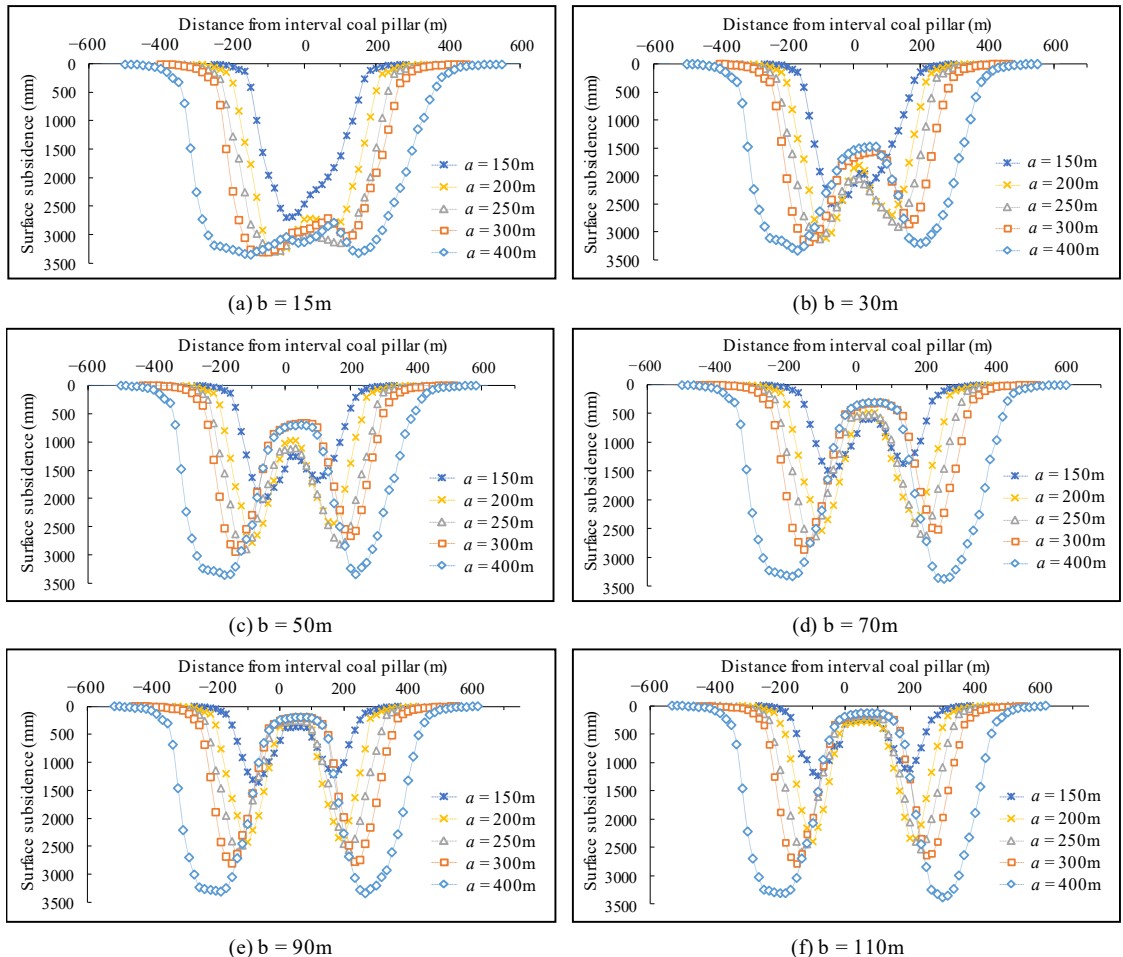

**Figure 7.** Surface subsidence curves of adjacent mining under different working face mining width and interval coal pillar width conditions.

It can be seen from Figure 7 that when the mining width of the working face is 150 m, 200 m, and 250 m, then the surface subsidence curve is steeper, the working face is in a state of insufficient mining, and the overlying rock structure has a strong control effect on the surface. A large number of voids and cracks exist between the caving zone and the fractured zone, and there is a large potential subsidence space on the surface. At this time, under the condition of adjacent mining, the width of the interval coal pillar has the greatest influence on the surface subsidence pattern. Affected by adjacent mining, the range of surface movement basin increases, the maximum subsidence value changes greatly, and the maximum subsidence position of the surface shifts to the side of the working face interval coal pillar. Taking the mining face width of 200 m as an example, the maximum surface subsidence value is 2321 mm, and the subsidence ratio is 0.44. When the width of the interval coal pillar is 30 m, the maximum subsidence value of the adjacent surface mining

is 3119 mm, which reaches the sufficient mining state. We observed that when the interval coal pillar is greater than 50 m, the maximum surface subsidence value decreases under adjacent mining conditions.

When the mining width of the working face reaches 300 m, then in that case the maximum surface subsidence value is approximately 2748 mm, which has not yet reached sufficient mining state. When the width of the coal pillar is small, it is greatly affected by the adjacent mining, and the surface subsidence of the two sides of the working face is considered "mutual enhancement". For example, the interval coal pillar is 30 m, and the maximum surface subsidence value after adjacent mining is 3181 mm. The maximum subsidence of the working face does not appear in the center of the goaf, but offsets a certain distance to the direction of the interval coal pillar.

When the mining width of the working face reaches 400 m, then the flat bottom of the surface subsidence basin appears, and the working face has reached a sufficient mining state. Under the influence of adjacent mining, the surface subsidence basin continues to expand; however, the maximum subsidence value is almost unchanged. At this time, there is no phenomenon that the maximum subsidence deviates from the center of the mining area. This observation suggests that the width of the interval coal pillar has little effect on the maximum subsidence value of the surface. The maximum subsidence value of the surface is 3310 mm when the 3104 working face is mined, and the maximum subsidence value of the surface after adjacent mining is 3339 mm.

At the same time, in order to further explore the influence of coal pillar width on adjacent mining, the surface subsidence curve above the coal pillar under different width–depth ratios is drawn, as shown in Figure 8.

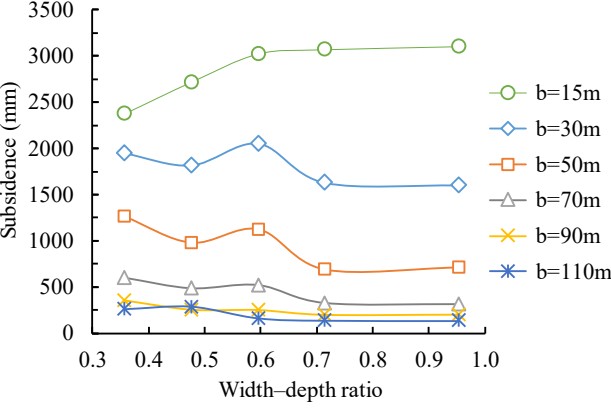

**Figure 8.** Surface subsidence value above the coal pillar under different width–depth ratios of adjacent mining.

Under the condition of different widths of interval coal pillars, the surface subsidence above the adjacent coal pillars changes with the width–depth ratio, which is mainly related to the bearing state of interval coal pillars. When the width of the coal pillar is 15 m, then the surface subsidence above the coal pillar increases first and then gradually stabilizes with the increase in the width of the mining face on both sides. At this time, the coal pillar is damaged by shear stress and is in a fully plastic state, the supporting effect disappears, and the surface subsidence is mainly affected by the change in mining face width.

When the width of the coal pillar is 30 m and 50 m, then, in that situation, the coal pillar plays a supporting role to a certain extent. With the increase in the width–depth ratio, the surface subsidence above the coal pillar experiences the process of "decrease–increase–decrease–stability". When the width–depth ratio of the working face is close to 0.6, the surface subsidence decreases. This is due to the fact that the adjacent mining surface basically reaches a sufficient mining state. The middle of the goaf is recompacted, and the weight of the coal pillar is changed from the weight of the overlying strata in the whole mining area to the weight of the overlying strata in a particular range near the bearing

coal pillar. This situation results in a smaller compression of the coal pillar. After that, the width–depth ratio increases again, and the stress above the coal pillar changes little and tends to gradually stabilize.

When the width of the coal pillar is 70 m, 90 m, and 110 m, then the elastic core area of the coal pillar is large enough, and the stable coal pillar is separated from the coordinated movement of the overlying strata on both sides, which weakens the influence of adjacent mining. At this time, only slight elastic compression deformation occurs in the interval coal pillar and overlying strata, and the surface subsidence value is small, less than 500 mm. In the width range of this coal pillar, with the increase in width–depth ratio, the surface subsidence above the coal pillar changes slightly and tends to be stable.

### 5.3. Analysis of Stress Change in Goaf and Coal Pillar during Adjacent Mining

Due to the existence of a certain width of the interval coal pillar between the working faces, the adjacent mining overlying strata structure is different from the single working face mining structure. In order to explore the variation law of overburden structure in adjacent mining structure, the stress distribution of goaf and interval coal pillar with the change of interval coal pillar width under different mining face width conditions is extracted, as shown in Figure 9. The original rock stress $\delta_0$ at the goaf and coal pillar is 8.9 MPa.

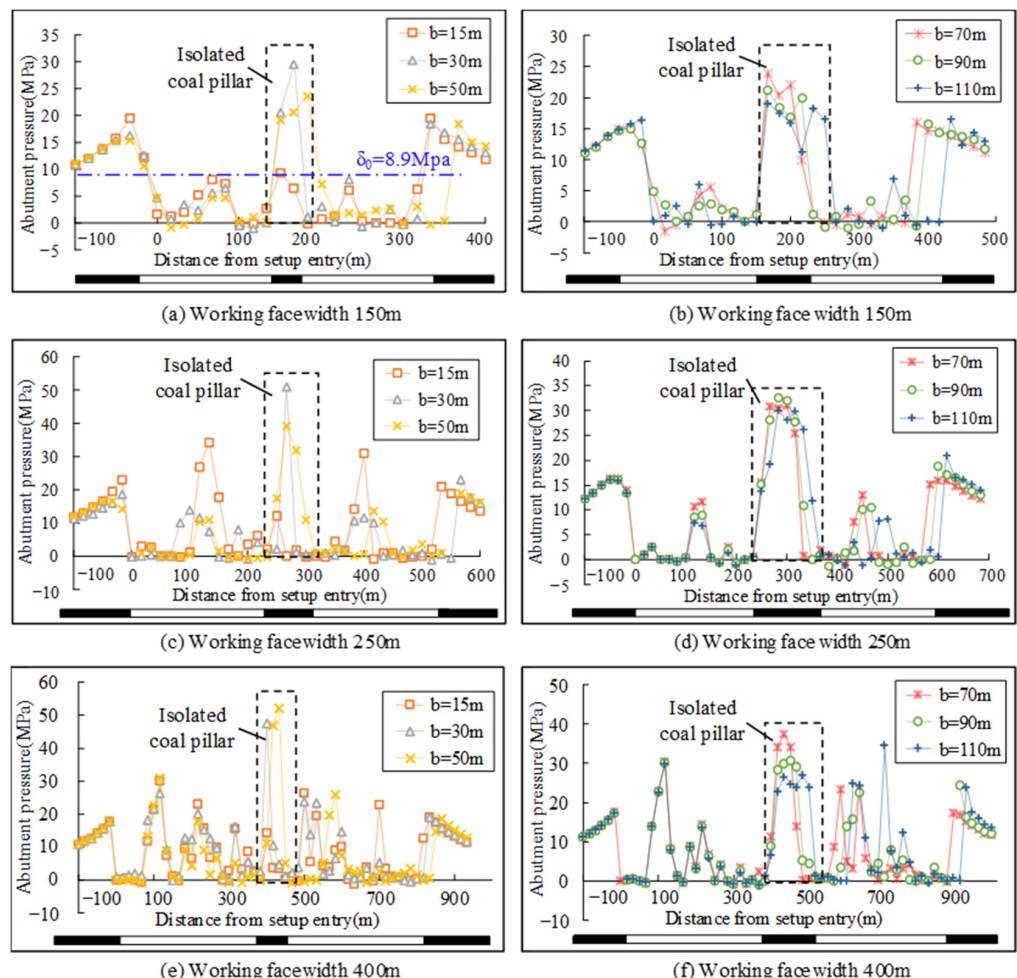

**Figure 9.** Abutment pressure distribution in goaf and coal pillar during adjacent mining of working face.

Similar to single-working face mining, adjacent mining forms stress concentration at both sides of the coal wall. In contrast, the stress distribution at the goaf and interval coal pillar shows different rules. From Figure 9, it can be seen that the peak stress of the

coal pillar is generally greater than that of the goaf during adjacent mining, and the peak stress of the goaf in the first mining face is slightly greater than that in the later mining face. Under the same working face width, the stress distribution law differs with different interval coal pillar widths. With the increase in coal pillar width, the peak stress of the interval coal pillar and goaf decreases. When the width of the coal pillar is large enough, then the stress of goaf on both sides is very close. In the stress distribution of the coal pillar, with the increase in coal pillar width, it gradually changes from "single hump" (one stress peak) to "double hump" (two stress peaks). In the coming discussion, the stress distribution of the goaf and coal pillar under the influence of adjacent mining is analyzed in detail for the working face in insufficient and sufficient mining.

When the width of the working face is 150 m (single working face is insufficient mining), then the stress distribution above the mining area and the interval coal pillar can be divided into three stages. In the first stage, when the width of the interval coal pillar is 15 m, then the coal pillar is a failure, and the overlying strata on both sides are connected to form a large mining face as a whole. However, it has not yet reached sufficient mining, and the peak stress exists on both sides of the coal wall. In the second stage, when the width of the coal pillar is 30 m, then the stress of the goaf on both sides is superimposed on the interval coal pillar. Furthermore, the stress distribution is a "single hump", and the stress peak reaches the maximum, which is 29.6 Mpa, and the stress concentration coefficient is 3.32. However, it should be noted that the stress values of the goaf on both sides are less than the original rock stress. In the third stage, when the interval coal pillar is 50 m, 70 m, 90 m, and 110 m, then the stress distribution of the interval coal pillar is a "double hump", and the peak stress gradually decreases.

When the width of the working face is 250 m (single working face is insufficient mining), then the stress distribution can still be divided into three stages. When the width of the interval coal pillar is 15 m, the coal pillar is a failure and cannot support the overlying strata, and the stress is transferred to the goaf. At this time, there are two stress peaks in the goaf. When the width of the coal pillar is 30 m and 50 m, the adjacent goafs on both sides affect each other, and the stress is superimposed on the interval coal pillar. The stress distribution is a "single hump", and the peak stress is 51.1 Mpa, and the stress concentration coefficient is 5.74. The stress in the goaf also increases, but it is still less than or slightly larger than the original rock stress. When the interval coal pillar is 70 m, 90 m, or 110 m, then the stress distribution of the interval coal pillar is a "double hump".

When the width of the working face is 400 m (a single working face is sufficient mining), in, this situation, there are many stress concentration phenomena above the mining area and interval coal pillar after adjacent mining. Overall, the greater the width of the interval coal pillar, the smaller the peak stress at the coal pillar and vice versa. When the width of the interval coal pillar is 15 m, the coal pillar is a failure, and there are many stress concentrations in the goaf. When the width of the interval coal pillar is 30 m, 50 m, and 70 m, then the stress of the coal pillar is a "single hump", and the peak stresses are 47.6 Mpa, 52.2 Mpa, and 34.2 Mpa, respectively. Furthermore, the stress concentration factors are 5.3, 5.8, and 3.8, respectively. Finally, when the interval coal pillar is 90 m and 110 m, then the stress of the coal pillar is a "double hump", and the peak stresses are (30.0 Mpa, 30.9 Mpa) and (26.6 Mpa, 27.1 Mpa), respectively.

## 6. Discussion

### 6.1. Comparison of "Space–Air–Ground" Integrated Monitoring Network with Existing Monitoring Methods

The geological environmental damage caused by coal mining has become a hot issue in current research. Especially in the western mining area, the mining face size is large, and the mining intensity is high. The strata and surface movement and deformation caused by coal mining are severe and the spread range is wide. Therefore, a fast and efficient settlement monitoring method is needed. However, as described in Section 2, the existing observation technologies have their own applicability requirements, advantages, and limi-

tations. For example, the traditional observation stations have high measurement accuracy, but they have various shortcomings such as small scale, high cost, and low efficiency. Similarly, UAV and InSAR technologies have problems such as low edge accuracy and decoherence in the application of mining deformation monitoring. Therefore, one of the technologies alone cannot effectively monitor the entirety of the surface subsidence basin information. A combination of various technologies might be more helpful in these scenarios. The "space–air–ground" integrated monitoring network combines the advantages of various monitoring technologies. For example, it uses UAV technology to observe a large deformation in the middle of the subsidence basin while it utilizes the InSAR mechanism to observe a small deformation at the edge of the subsidence basin. Furthermore, GNSS technology is also used to effectively supplement and verify the accuracy of the obtained results and outcomes. In addition to these, the multi-source monitoring data fusion and aggregation methods within the framework of IoT technology is more helpful for obtaining surface subsidence information more economically, efficiently, and accurately.

In this paper, the "space–air–ground" integrated monitoring network was applied in the study area and an IoT-based framework was proposed. The overall mean square error of the InSAR and UAV fusion subsidence basin was calculated as approximately 33.2 mm, and the mean square error of the discrete element numerical simulation and the fusion subsidence result was 13.4 mm. The relative error is as low as 3.8%, which can meet the monitoring requirements of coal mining subsidence damage and can provide more reliable data support for discrete element simulation research.

Zhou et al. [9] used UAV photogrammetry to monitor the surface subsidence of coalmine areas. The root mean square error (RMSE) of the subsidence basin obtained was 81 mm. Due to the limitation of UAV monitoring accuracy, it is impossible to describe the edge part of the subsidence basin. Different from the current monitoring methods, this paper combines InSAR technology and can effectively monitor the small deformation area of the edge. Many scholars have mainly used InSAR technology to monitor and analyze the surface deformation in the mining area [13,16]. However, for the high-intensity mining conditions in the western mining area, the central deformation of the subsidence basin is large and the phase decoherence is serious. Different from the current monitoring methods, this paper combines UAV and InSAR technology, and uses the GNSS technology to supplement and verify the "space–air" monitoring, which can obtain the subsidence basin comprehensively and accurately. Apart from these, the relevant literature deliberated in Section 2 has not introduced IoT technology, which provides a platform for data preprocessing, aggregation, fusion, and automated processing.

It is worth noting that the threshold of the observation ranges of UAV and InSAR in this method has a great influence on the data fusion results. If the difference between the two thresholds is large, then it means that the effective monitoring data available in the middle part of the probability integral method is reduced, and the accuracy of the fused data also decreases. In addition, due to the limitations of field working conditions, the number of measuring points selected for data accuracy evaluation is relatively small, which may lead to slightly higher accuracy evaluation results.

### 6.2. Surface Sufficiency Change and Interval Coal Pillar Stability in Adjacent Mining

The relationship between the width–depth ratio of the working face and the surface subsidence ratio in the study area was analyzed, and the Boltzmann fitting curve is shown in Figure 10.

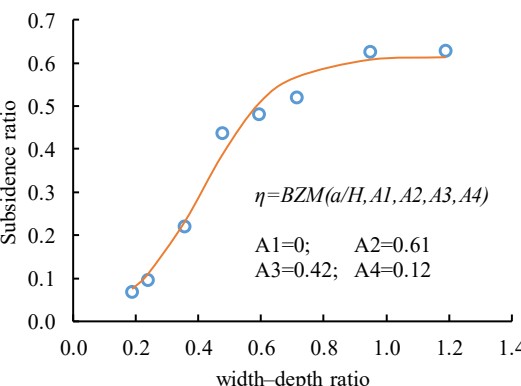

**Figure 10.** Boltzmann fitting curve of surface subsidence ratio.

According to the simulation results, the inversion parameters were obtained using least square fitting: $A_1 = 0$; $A_2 = 0.6138$; $A_3 = 0.4184$; $A_4 = 0.1168$. The subsidence ratio and width–depth ratio of the mining area satisfy the Boltzmann function as provided in Formula (3):

$$\eta = 0.61 - \frac{0.61}{1 + e^{(a/H - 0.42)/0.12}}, \quad R^2 = 0.95 \tag{3}$$

where $\eta$ is the surface subsidence ratio; $a$ is the width of the working face; and $H$ is the mining depth of working face. The width-to-depth ratio is positively correlated with the subsidence ratio. With the increase in the width-to-depth ratio, the surface subsidence first increases rapidly, then increase slowly, and finally reaches the maximum value. When the width–depth ratio is greater than 0.42, then the subsidence rate increases slowly with the increase in the width–depth ratio. When the width–depth ratio is less than 0.42, then the subsidence ratio increases rapidly.

After the adjacent working face is mined, then the surface sufficiency and the maximum subsidence change rapidly. At this time, the parameter $a$ in Formula (3) should be the equivalent mining width, characterized by $a'$ for the two adjacent mining faces, and the calculation formula is provided in Equation (4):

$$a' = a + ka \tag{4}$$

In the above formula, $k$ is the mining width reduction coefficient affected by the interval coal pillar, and the value range is (0, 1). The value is related to the width of the coal pillar and its mechanical properties. When $k = 0$, then it means that the coal pillar is completely isolated from the mining influence of the working faces on both sides. When $k = 1$, then it means that the interval coal pillar fails.

The maximum subsidence increment of the adjacent mining surface reflects the disturbance degree of the adjacent working face. The maximum surface subsidence value after adjacent mining is brought into Formula (3), the equivalent mining width $a'$ is inversely deduced, and the reduction coefficient k under the condition of each interval coal pillar width is fitted, as shown in Figure 11.

It can be seen from Figure 11 that the length reduction coefficient of the adjacent mining face in the mining area is $k = 2.954e^{-0.052b}$. When $k = 1$, the critical failure width of the coal pillar $b_0 = 20.5$ m, which is similar to the calculation result of 24.3 m using the Wilson formula and cusp catastrophe theory in Reference [38]. It is determined that the width of the interval coal pillar in the isolated goaf and the maintenance roadway should be greater than 20.5 m under the adjacent mining conditions of the mine.

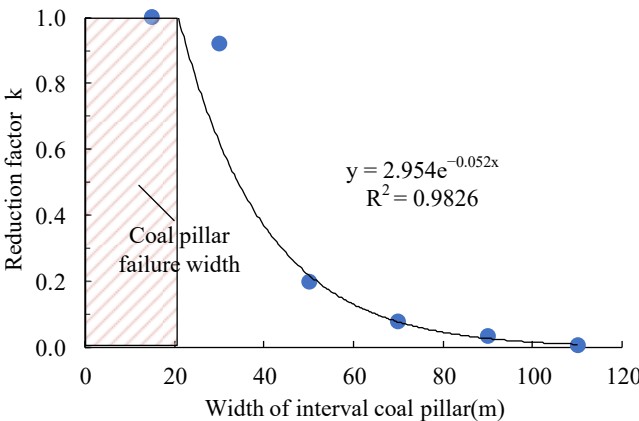

**Figure 11.** Relationship between length reduction coefficient of working face and width of interval coal pillar in adjacent mining.

### 6.3. Evolution Law of Overlying Strata Structure in Adjacent Mining Face

The overburden structure of a working face mining mainly includes two scales: arch and beam. The pressure arch belongs to the whole structure, which plays a macro supporting role in the whole overlying strata. The key stratum existing in the overlying strata belongs to the stratum structure, which controls the movement of the soft rock and surface in the upper part of the stratum [21,22,29]. According to the theory of pressure arch, the high-stress trace of mining overburden is arched; the arch foot shows obvious stress concentration, as shown in Figure 12; and the maximum vertical stress at the arch foot is generally greater than 1.5 times the original rock stress [1,39]. The main key stratum controls surface movement, and its structural characteristics are directly related to the surface subsidence characteristics. Therefore, based on the "arch–beam" theory, combined with the distribution of mining area stress and surface subsidence, this paper analyzes the evolution law of overlying strata structure in the stope.

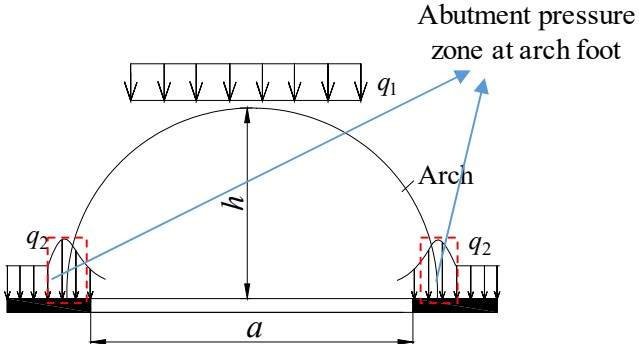

**Figure 12.** Vertical stress distribution of overburden arch structure.

The high-stress area of the goaf and coal pillar is a necessary condition for the formation of the arch foot. The distribution of stress peak (where the vertical stress is greater than 1.5 times the original rock stress) in the goaf and coal pillar of each simulation scheme is shown in Table 4.

**Table 4.** Stress distribution of goaf and coal pillar under adjacent mining conditions.

| Width of Working Face | Width of Interval Coal Pillar | Sufficiency State of Single Working Face | Coal Pillar Stability State | Number of Peak Stress | |
|---|---|---|---|---|---|
| | | | | Position of Interval Coal Pillar | Goaf |
| $a_1 = 150$ m | $b = 15$ m | Insufficient | Instability | 0 | 0 |
| | $b = 30$ m | Insufficient | Stability | 1 | 0 |
| | $b = 50$ m, 70 m, 90 m, 110 m | Insufficient | Stability | 2 | 0 |
| $a_2 = 200, 250, 300$ m | $b = 15$ m | Insufficient | Instability | 0 | 2 |
| | $b = 30$ m, 50 m | Insufficient | Stability | 1 | 0 |
| | $b = 70$ m, 90 m, 110 m | Insufficient | Stability | 2 | 0 |
| $a_3 = 400$ m | $b = 15$ m | Sufficient | Instability | 0 | 2 |
| | $b = 30$ m, 50 m, 70 m | Sufficient | Stability | 1 | 4 |
| | $b = 90$ m, 110 m | Sufficient | Stability | 2 | 4 |

Adjacent mining is affected by the width of the working face and the width of the interval coal pillar. The mining width of the working face determines the subsidence potential of the initial mining face, the surface sufficiency state after adjacent mining, and the width of the interval coal pillar determines the mutual influence degree of adjacent mining. In the study mining area, under the condition of working face mining width of 150 m–400 m, according to the insufficient (the main arch height does not exceed the surface) and the sufficient mining state, the mining width is grouped into $a_1$ (150 m), $a_2$ (200 m, 250 m, 300 m), and $a_3$ (400 m). The adjacent mining is affected by the adjacent mining sufficiency and the interval coal pillar, which is mainly manifested as the evolution form of the overburden structure of the arch and the hinge state of the key block. We can summarize our major findings, as discussed below:

- For mining widths $a_1$ and $a_2$, there is a main arch structure in the overlying strata. When the interval coal pillar width $b \geq b_0$, then the coal pillar is stable, the goaf on both sides is separated, and the double arch feet are formed above the coal pillar. The development height of the main rock vault on both sides does not exceed the surface, forming the structure of the double main arch + "W-shaped" beam structure, as shown in Figure 13c. At this time, if the width of the coal pillar is relatively small ($b = 30$ m or 50 m), then the two arch feet overlap, and the single stress peak state appears above the coal pillar, and the vertical stress is larger. When the interval coal pillar width $b < b_0$, the coal pillar is affected by the concentrated stress, and failure occurs. The arch structure of the two working faces is connected. If the mining face width is small ($a_1$), then the height of the arch in the adjacent mining of the working face is still less than the mining depth, that is, $h_m(a') \leq H$. Furthermore, the height of the connected main arch does not exceed the ground, forming a single main arch + "V-shaped" beam structure, as shown in Figure 13a. When the width of the mining face is large ($a_2$), and the arch height of the adjacent mining of the working face is greater than the mining depth, that is, $h_m(a') > H$, then the connected main rock arch exceeds the ground, and two vice rock arches are formed on both sides, forming a double vice arch + "U-shaped" beam structure, as shown in Figure 13b.

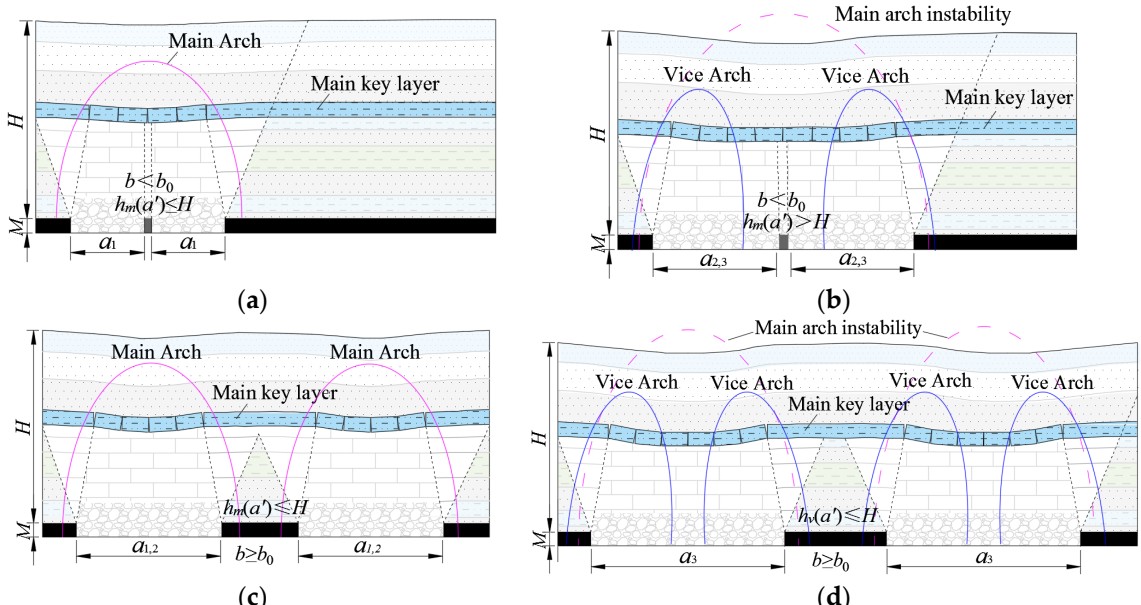

**Figure 13.** Working face adjacent mining overlying strata "arch–beam" structure model. (**a**) single main arch + "V-shaped" beam; (**b**) double vice arch + "U-shaped" beam; (**c**) double main arch + "W-shaped" beam; (**d**) four vice arches + double "U-shaped" beams.

- For mining width $a_3$, the overlying strata has a vice arch structure. When the width of the interval coal pillar $b < b_0$, then the coal pillar is a failure, and the overlying strata structure on both sides of the working face is connected to form the structure of the double vice arch + "U-shaped" beam. At this time, there is no obvious stress concentration in the center of the goaf, and only several original rock stress recovery areas appear, as shown in Figure 13b. When the width of the interval coal pillar $b \geq b_0$, then the coal pillar is stable, separated from the goaf on both sides, and a double arch foot is formed above the coal pillar (the arch foot is shared when the coal pillar is small). There are vice arches on both sides of the coal wall, forming a structure of four vice arches + double "U-shaped" beams, as shown in Figure 13d.

In addition, if $h_v(a') > H$, that is, the height of the vice arch exceeds the surface, then, in that situation, there is no arch bearing in the overlying strata above the working face, and the surface damage is severe. This situation generally exists in mining areas where the mining depth–mining thickness ratio is less than 30.

In this study, due to the limitation of research methods, it is assumed that the length of the broken blocks of the main key strata is the same, and the influence of factors such as mining sequence and boundary conditions is not considered. It is impossible to explore the relationship between the breaking length and the surface subsidence skewness. These need to be further studied in the future research.

## 7. Conclusions and Future Work

In this paper, based on the advantages of various subsidence monitoring technologies and the technical framework of the Internet of Things (IoT), a "space–air–ground" integrated collaborative monitoring network is constructed. The evolution law of overlying strata structure is studied based on discrete element simulations and theoretical analysis. Furthermore, a discrete element mechanical parameter inversion method based on the "space–air–ground" integrated monitoring network is proposed. Using numerical simulations and real datasets, the main results are summarized as follows:

- Combined with the threshold of surface subsidence monitored using UAV and InSAR, a monitoring method based on "space–air–ground" network integration is proposed. The proposed method can make up for disadvantages such as the UAV not being

able to effectively monitor the small deformation at the edge of the subsidence basin and the InSAR not being able to monitor the large deformation at the center of the subsidence basin. In addition, Internet of Things technology can fuse and aggregate the data. The overall mean square error of the InSAR and UAV fusion subsidence basin is 33.2 mm, and the mean square error of the discrete element numerical simulation and fusion subsidence results is 13.4 mm. Based on the monitoring network, a discrete element mechanical parameter inversion method was proposed.

- A discrete element numerical simulation model for adjacent mining in a thick coal seam working face was also established. The surface subsidence law of adjacent mining under different working face widths and interval coal pillar widths was revealed. The Bozeman function model of the surface subsidence ratio changing with the aspect ratio was inverted, $\eta = 0.61 - \frac{0.61}{1+e^{(a/H-0.42)/0.12}}$. The calculation formula for the width reduction coefficient of the adjacent mining face was established, $k = 2.954e^{-0.052b}$, and the critical failure width of the coal pillar was calculated as 20.5 m. Therefore, in order to ensure safety, the width of the interval coal pillar should be greater than this size when the adjacent mining face is laid in this study area.

- Based on the theory of arch and beam structure and the results of numerical simulations, the structural model of adjacent mining overlying strata with different mining widths and interval coal pillar widths in the mining area was constructed. These include a single main arch + "V-shaped" beam, double vice arch + "U-shaped" beam, double main arch + "W-shaped" beam, and four vice arch + double "U-shaped" beam.

In the future, we intend to explore the relationship between breaking length and surface subsidence skewness, particularly when the length of the broken blocks of the main key strata is different. Moreover, we will also study the influence of other factors such as mining sequence and boundary conditions under various conditions and parameters. In addition to these, we will also investigate the impact of the data aggregation mechanism on the accuracy of the proposed "space–air–ground" integrated collaborative monitoring network.

**Author Contributions:** Conceptualization, Y.Z. (Yuanhao Zhu); methodology, J.K. and W.Z.; formal analysis, Y.Y.; investigation, Y.Z. (Yanjun Zhang); resources, A.D.; writing—original draft, Y.Z. (Yuanhao Zhu). All authors have read and agreed to the published version of the manuscript.

**Funding:** This research was funded by the National Natural Science Foundation of China, grant number (51574242, Y.Y.); the Fundamental Research Funds for the Central Universities, grant number (2022YJSDC20, 2022YJSDC19, Y.Y.). We thank all the reviewers for their valuable comments.

**Conflicts of Interest:** The authors declare that they have no competing interests.

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
