# Peer review of "Study on the Evolution Law of Overlying Strata Structure in Stope Based on “Space–Air–Ground” Integrated Monitoring Network and Discrete Element"

_drones, doi:10.3390/drones7050309_

Round 1
Reviewer 1 Report
The topic presented in the paper is important and on time. In the reviewed paper based on the advantages of various subsidence monitoring technologies and the technical framework of the Internet of Things, a “space-air-ground” integrated collaborative monitoring network is constructed. The evolution law of overlying strata structure is studied based on discrete element simulation and theoretical analysis. In my opinion, the paper can be published, after taking into account the following remarks:
- in the abstract section, the Authors presented the results of their research in too much detail. Submitting details in the abstract is not required. In general, the obtained results should be specified, along with an indication of their further use,
- at the end of the literature review section (i.e. second section) some sentences with conclusions from state in the art in this area should be provided,
- the structure of the article is not appropriate. Some sections are divided into sub-sections and sub-subsections and their content is too short, e.g. "3.1.2. InSAR solution of subsidence basin". The Authors should develop the content of each sub-subsections or just not divide subsections into further sub-subsections,
- the background of discrete element numerical simulation like assumptions, limitations, and requirements are lacking. It should be added,
- there is a lack of explanation for some used variables/acronyms in the equations, e.g. like in equation (3). It should be improved,
- the Conclusions section is written in a very general way and should be extended by adding some detailed conclusions stated based on the results presented in the paper.
Reviewer 2 Report
This paper focuses on the geological environment damage caused by coal mining in the western mining area, where the mining intensity is high and the working face is usually large, leading to complex overlying strata structure evolution and intense surface movement and deformation. To monitor the surface effectively and study the overlying strata structure of the stope, the authors propose a "space-air-ground" integrated collaborative monitoring network that combines various subsidence monitoring technologies and the technical framework of IoT. The authors also conduct discrete element simulation and theoretical analysis to reveal the surface subsidence law and overlying strata structure model of adjacent mining in the mining area. The research results can provide technical support and theoretical reference for mining damage monitoring, subsidence control, and prediction in western mines.
The paper is interesting and flows well. It is well written and the results herein presented seem solid. However, here are some aspects the authors could consider to take into account in order to improve the quality of the manuscript:
- The study of the related literature is not very easy to grasp. I suggest the authors to add a section contain the related literature and a more clear comparison of the proposed approach w.r.t. the approaches found in literature.
- As for the related approaches, the authors could consider citing [doi.org/10.1016/j.compeleceng.2021.107572]. In fact, the approach proposed by the authors could easily be used to enhance the work presented in this reference.
- A more detailed discussion of the limitations of this approach could be provided.
- Some typos are present; I suggest the authors to carefully read the manuscript and fix them.
Reviewer 3 Report
Dear Authors,
The language used in the manuscript needs to be improved as it is very difficult or even impossible to understand several sentences. It is also unclear to comprehend how the IoT was involved in the study. The methodology was not sufficiently explained for photogrammetric and InSAR processing.
Therefore, I cannot recommend publication of the manuscript.
The quality of presentation is unfortunately very low.
Round 2
Reviewer 2 Report
The authors successfully addressed my concerns.
Reviewer 3 Report
Dear Authors,
Thank you for the revisions. I see that the language has been improved. My main concern is still about IoT. To my knowledge, the term IoT identifies dynamic systems running over networks. I don't see any dynamic systems in the paper, but an integration of multi-source data (UAV and InSAR) through a simple spatial overlay and DEM subtraction. This point needs to be clarified before the manuscript can be published.
In addition, I noticed errors with heading numbers.
Kind regards
Minor improvements may be needed.
